The Significance of the preoperative lactate dehydrogenase/albumin Ratio in the Prognosis of Colon Cancer: a retrospective study

Xie Zhihui 1
Zhou Hui 2
Wang Lei 1 xzhihui@yeah.net
Wu Yibo 2 moliaty@aliyun.com
1 Department of General Surgery, Affiliated Hospital of Jiangnan University , Wuxi, Jiangsu , China
2 Human Reproductive Medicine Center, Affiliated Hospital of Jiangnan University , Wuxi, Jiangsu , China
Zhang Jian
Electronic publication date: 2022 Mar 11
Publication date: 2022
Volume: 10
Electronic Location ID: e13091
Received 2021 Jul 21; Accepted 2022 Feb 18
Copyright: © 2022 Xie et al.
Copyright year: 2022
Copyright holder: Xie et al.
License: This is an open access article distributed under the terms of the Creative Commons Attribution License, which permits unrestricted use, distribution, reproduction and adaptation in any medium and for any purpose provided that it is properly attributed. For attribution, the original author(s), title, publication source (PeerJ) and either DOI or URL of the article must be cited.
License URL: https://creativecommons.org/licenses/by/4.0/

Keywords: Colon cancer, Prognosis, lactate dehydrogenase, Albumin

Funding: National Natural Science Fund 81701511 Young and Middle-Aged People of Wuxi Health Committee BJ2020047 Wuxi Health Commission Q201926 This study was supported by a grant from the National Natural Science Fund (81701511), and grants from the Top Talent Support Program for Young and Middle-Aged People of Wuxi Health Committee (BJ2020047) and the Young Fund of Wuxi Health Commission (Q201926). The funders had no role in study design, data collection and analysis, decision to publish, or preparation of the manuscript.

==============================
Background

We explored the relationship between the platelet-lymphocyte ratio (PLR), the prognostic nutritional index (PNI), the lactate dehydrogenase-albumin ratio (LDH/albumin ratio; LAR), the controlling nutritional status (CONUT) score, and the long-term survival of colon cancer patients.

Methods

We conducted a retrospective analysis of the clinical data and follow-up materials of 126 patients with colon cancer who underwent surgical treatment in the Department of Gastrointestinal Surgery of Jiangnan University Affiliated Hospital from June 2012 to December 2015. The receiver operating characteristic curve (ROC) was used to distinguish the high ratio group from the low ratio group. The Kaplan Meier method was used to draw the survival curve in our survival analysis. The log rank test was used for the univariate analysis and the Cox multivariate regression analysis was used to analyze the correlation between preoperative PLR, PNI, LAR, conut scores, and overall survival (OS) and progression free survival (PFS) of patients with colon cancer.

Results

The median follow-up time was 72 months. The OS rates at 3 and 5 years were 83.3% and 78.5%, respectively. The PFS rates at 3 and 5 years were 79.3% and 77.6%, respectively. The 3-year and 5-year OS rates in the low LAR group (≤4.91) were 90.9% and 87.1%, respectively, and were 56.0% and 44.0% in the high LAR group (>4.91) respectively. Univariate and multivariate analyses showed that the LAR value was correlated with OS and PFS (P < 0.05).

Conclusion

A high preoperative LAR is an independent predictor of the prognosis of colon cancer patients.

Introduction

Colon cancer is one of the most common malignant tumors of the digestive tract. Epidemiological data have shown that colon cancer ranks second in the incidence of malignant tumors in the United States and Europe (Matuchansky, 2017). In China, the incidence of colon cancer ranks fourth in terms of malignant tumors and fifth in terms of mortality (Chen et al., 2013). In recent years, the incidence and mortality rate of colon cancer have both increased year-by-year, which poses a serious threat to human health (Zeng et al., 2016). Although advances in both radical surgery and comprehensive treatment methods have improved the survival rate of patients with colon cancer, some cases progress to distant metastasis, which has a significantly lower survival rate.

The TNM (tumor-node-metastasis) staging system, traditionally used in clinical practice, is an important way to determine the prognosis of colon cancer, however, we found that clinical outcomes of the patients with the same TNM stage vary from person to person, due to the deficiency of individualized assessment, so our study considers the combination of effective biochemical indicators to analyze their effects on the prognosis of colon cancer patients. At present, it is believed that a patient’s inflammatory response and immune status are related to their therapeutic response and cancer prognosis. The inflammatory response of tumors and the protective immunity of the host are interrelated as a dynamic process, and inflammation can alter this dynamic balance to promote tumorigenesis (Elinav et al., 2013; Mantovani et al., 2008). Among the dynamic observations representing immune status and the inflammatory response, the platelet-to-lymphocytes ratio (platelets-lymphocytes ratio PLR), the nutritional prognosis index (prognostic nutritional index, PNI), the control nutritional status (controlling status, CONUT) score, and the lactic acid dehydrogenase vs albumin ratio (LDH/albuminin, LAR) have been used in a variety of cancer prognosis studies (Dan & Lianyii, 2020; Liang et al., 2017; Mirili et al., 2019; Smith et al., 2016). It has been reported that the PLR, PNI, and CONUT scores can affect colon cancer survival prognosis, but it is still disputed whether they can be used as independent prognosis factors (Iseki et al., 2015; Suzuki et al., 2018; Weizhong, 2017). The prognosis value of the LAR value in colon cancer has not been studied in China. Therefore, we retrospectively analyzed the relationship between preoperative PLR, PNI, LAR, and CONUT scores and the overall survival (OS) and progression-free survival (PFS) of 126 patients with colon cancer to identify the best independent prognostic factors to help effectively layer risk and improve personalized treatment options.

Materials and Methods

General information

We collected data on colon cancer patients who were initially treated for gastrointestinal surgeries at the Affiliated Hospital of Jiangnan University between June 2012 and December 2015. Each patient’s cancer stage was determined using the 2012 International Anti-Cancer Alliance (UICC) 7th Edition Phased Standard with the following criteria: (1) patients with pathologically confirmed primary colon cancer; (2) patients whose serum biochemical examination was performed within 1 week of surgery and had complete data; and (3) patients who underwent radical surgery. The exclusion criteria were as follows: (1) the presence of other primary tumors; (2) patients with acute or chronic inflammatory diseases or infections before surgery; (3) patients with an intestinal obstruction or perforation; (4) patients with severe heart, lung, and kidney diseases; (5) patients who had neoadjuvant treatments such as radiotherapy and chemotherapy performed before surgery. This study was approved by the Medical Ethics Committee of the Affiliated Hospital of Jiangnan University (JNMS01201800138). The data for this retrospective study are anonymous, and the requirement for informed consent was therefore waived.

Methods

The serum biochemical examination for all patients was performed within 1 week before surgery and the data were complete. The hospital used the SYSMEX blood cell analyzer for blood analysis, and the Roche cobas c702 biochemical instrument to detect liver function and collect laboratory data including platelets, lymphocytes, albumin, cholesterol, and lactic acid dehydrogenase. Treatment of these patients was informed by the 2012 1st edition of the NCCN guidelines: postoperative follow-up observation was used for stage I patients; stage II patients with low-risk factors were treated with capecitabine; patients with high risk factors were treated with 6–8 courses of mFOLFOX adjuvant chemotherapy; and stage III and IV patients were treated with 6–8 courses of mFOLFOX adjuvant chemotherapy. Clinical data, including the patient’s age, gender, tumor site, and TNM stage, were collected from the patient’s disease-history record and pathology report. The laboratory data formula was calculated as follows:

PNI-Serum albumin concentration (g/L) + 5× lymphocyte count (×109/L) PLR-Platelet count (×109/L)/lymphocyte count (×109/L)

LAR-Lactic acid dehydrogenase(U/L)/albumin (g/L)

Follow-up

The follow-up of survivors was carried out by outpatient reexamination, in-patient reexamination, and telephone and e-mail interviews. The progress follow-up was comprised of a full-body physical examination, an analysis of tumor markers, and a computed tomography (CT) or magnetic resonance imagery (MRI) of the head, chest, abdomen, and pelvis. Follow-up was performed every 3 months for 3 years, and then every 6 months for more than 3 years after surgery. Follow-up was performed annually until December 2020. OS was defined as the time from the day of surgery to the time of death, or until the end of the study. PFS was defined as the time between postoperative pathological diagnosis and a physical examination suggesting the progression to distant metastasis.

Statistical methods

R software was used to calculate the cutoff value of the PLR, PNI, and LAR, and SPSS 23.0 software was used for the statistical analysis. The Kaplan–Meier method was used to calculate OS and PFS, and the log rank test and the Cox proportional hazards regression model were used for the multivariate analysis. P-values < 0.05 were considered statistically significant.

Results

Using PLR, PNI, and LAR to determine the best cut-off value of OS

The ROC curve was used to calculate the Youden index (sensitivity + specificity − 1), with patient death as the endpoint. As shown in Table 1, the area under the curve (AUC) of the PLR, PNI, and LAR were 0.582, 0.520, and 0.754 respectively, and their optimal cutoff values were 196.5, 46.2, and 4.9109 respectively. The optimal specificity and sensitivity were as follows: PLR, 43.3% and 78.1%; PNI, 56.7% and 56.2%; and LAR, 50.0% and 90.6% (Fig. 1).

Figure 1 Best cutoff value of preoperative PLR, PNI, and LAR of 126 colon cancer patients.

Table 1 Critical values of PLR, PNI, LAR.

Index	AUC	Sensitivity	Specificity	Jorden index	Cutoff value	
PLR	0.588	42.1	75.9	0.1803	>193.64	
PNI	0.544	23.8	90.7	0.1455	≤39.4	
LAR	0.722	47.6	90.7	0.3836	>4.9109	

CONUT scoring criteria and grouping

The 126 patients with colon cancer were divided into the following three groups according to their nutritional status: a CONUT1 (normal) group (CONUT: 0–1 point, n = 57), a CONUT2 (mild malnutrition) group (CONUT: 2–4 points, n = 56), and a CONUT3 (moderate-to-severe malnutrition) group (CONUT: 5–12 points, n = 13). CONUT Scoring Criteria was shown in Table 2.

Table 2 CONUT scoring criteria.

Indicators	Normal	Mild	Moderate	Severe	
Serum albumin (g/dl)	≥3.50	3.00–3.49	2.50–2.99	<2.50	
Scoring	0	2	4	6	
Total lymphocyte count/(nm3)	≥1600	1,200–1,599	800–1,199	<800	
Scoring	0	1	2	3	
Total cholesterol (mg/dl)	>180	140–179	100–139	<100	
Scoring	0	1	2	3	
CONUT score	0-1	2–4	5–8	9–12	

Clinical characteristics of patients

The clinical characteristics of 126 patients with colon cancer are shown in Table 3. A total of 126 colon cancer patients (66 males and 60 females) with a median age of 66 years (range: 19–89 years) were admitted for surgical treatment. In 2012, among these 126 patients, there were nine cases of stage I cancer, 48 cases of stage II cancer, 57 cases of stage III cancer, and 12 cases of stage IVA cancer with cancer stages defined by the 7th edition of the Union for International Cancer Control staging criteria. A median number of follow-up time of the 126 patients, who have been in touch with our research team, is 72 months (range: 2–101 months). The correlation between the LAR and clinical pathological characteristics is shown in Table 3. The LAR was significantly associated with M-phased (P < 0.002) and CONUT scores (P < 0.002), but was not statistically significant in correlation with age, gender, T-stage, N staging, clinical staging, PNI, and PLR.

Table 3 Clinical characteristics of 126 patients with colon cancer.

Variable	Overall
(cases (%))	High LAR group (cases (%))	Low LAR group (cases (%))	P value	
Total	126	25	101		
Age (years)				0.617	
≤66	65	10	46		
>66	61	15	55		
Sex				0.624	
Male	66	12	54		
Female	60	13	47		
T stage				0.122	
T1–T2	12	0	12		
T3–T4	114	25	89		
N stage				0.394	
N0	60	10	50		
N1-2	66	15	51		
M stage				0.002	
M0	114	18	96		
M1	12	7	5		
TNM stage				0.137	
I-II	57	8	49		
III-IV	69	17	52		
PNI				0.140	
≤46.2	59	15	44		
>46.2	67	10	57		
PLR				0.102	
≤196.5	92	15	77		
>196.5	34	10	24		
CONUT				0.000	
CONUT normal	57	6	51		
CONUT mild	56	8	48		
CONUT moderate severe	13	11	2		

Prognosis and survival analysis of patients with colon cancer

The median follow-up time of the patients was 72 months, with OS rates of 83.3% and 78.5% and PFS rates of 79.3% and 77.6% at 3 and 5 years, respectively. The patients were divided into high fraction groups and low fraction groups based on the best critical value. The OS rates in the low LAR group at 3 and 5 years were 90.9% and 87.1%, and those in the high LAR group were 56.0% and 44.0%, respectively.

A univariate analysis showed that gender, N stage, M stage, clinical stage, PLR, and LAR were significantly associated with PFS in patients with colon cancer (P < 0.01) (see Table 4). In addition, N stage, M stage, clinical stage, PLR, and LAR were also statistically correlated with OS (P < 0.01). The OS and PFS of the high LAR group (LAR > 4.91) were lower than those in the group with low LAR (LAR ≤ 4.91), representing a negative correlation (P < 0.01). To better evaluate the prognostic indicators, the gender, N stage, M stage, clinical stage, PLR, and LAR for each patient were included in the Cox multivariate regression model of PFS for analysis. The results showed that gender, clinical stage and LAR were all associated with PFS in patients with colon cancer (P < 0.01) (Fig. 2). In addition, the N stage, M stage, clinical stage, PLR, and LAR of each patient were included in the Cox multi-factor regression model of OS. The M stage, clinical stage, PLR, and LAR were found to be associated with OS in patients with colon cancer (P < 0.05) (Fig. 3).

Table 4 Univariate and multivariate analysis of OS and PFS in colon cancer patients.

Item	Univariate analysis of PFS	Multivariate analysis of PFS	Univariate analysis of OS	Multivariate analysis of OS	
	HR (95% CI)	P	HR (95% CI)	P	HR (95% CI)	P	HR (95% CI)	P	
Gender	3.203 [1.410–7.278]	0.005	3.158 [1.359–7.339]	0.008	1.423 [0.694–2.916]	0.336			
Age	0.809 [0.386–1.698]	0.576			1.405 [0.668–2.952]	0.370			
T stage	3.041 [0.413–22.382]	0.275			1.585 [0.378–6.655]	0.529			
N stage	9.220 [2.781–30.565]	0.000			3.528 [1.513–8.227]	0.004			
M stage	7.795 [3.470–17.506]	0.000			9.086 [4.034–20.465]	0.000	3.527 [1.270–8.352]	0.014	
Clinical stage	27.898 [3.788–205.444]	0.001	25.527 [3.443–189.267]	0.002	5.088 [1.946–13.303]	0.001	3.943 [1.440–10.798]	0.008	
PNI	1.355 [0.635–2.893]	0.432			0.638 [0.310–1.315]	0.223			
PLR	2.831 [1.345–5.955]	0.006	2.025 [0.930–4.411]	0.076	2.634 [1.278–5.430]	0.009	2.647 [1.255–5.581]	0.011	
Conut	1.196 [0.691–2.069]	0.523			1.618 [0.963–2.721]	0.069			
LAR	4.532 [2.148–9.562]	0.000	3.643 [1.667–7.965]	0.001	6.112 [2.955–12.643]	0.000	4.162 [1.851–9.359]	0.001	

Figure 2 PFS curve of preoperative PLR, PNI, and LAR scores of 126 colon cancer patients.

(A) PNI for PFS; (B) PLR for PFS; (C) CONUT scores for PFS; (D) LAR for PFS.

Figure 3 Survival curve of preoperative PLR, PNI, and LAR scores of 126 colon cancer patients.

(A) PNI for OS; (B) PLR for OS; (C) CONUT scores for OS; (D) LAR for OS.

Discussion

In this study, the PLR, PNI, LAR, and CONUT score, which are four observation indices representing inflammatory response and immune status, were compared in terms of their ability to determine the prognosis of colon cancer patients. The OS and PFS of patients in the high LAR group were significantly lower than those in the low LAR group. Furthermore, a multivariate analysis showed that the LAR was negatively correlated with the OS and PFS of patients with colon cancer, which was statistically significant. The LAR combines two important indexes: LDH and albumin. During cell metabolism, LDH is a key enzyme that is required for the conversion of pyruvate to lactic acid. Tumor cells proliferate abnormally, consuming large quantities of oxygen in their microenvironment, and rely on anaerobic glycolysis to provide energy to maintain their survival and proliferation. It has been reported that the serum concentration of LDH is reflective of the tumor burden and the degree of hypoxia (Smith et al., 2016). Serum albumin is an effective index that reflects the systemic nutritional status of patients with cancer, and poor nutritional status is usually associated with poor tumor prognosis (Saito et al., 2018). Albumin is considered an important marker for cancer control by stabilizing cell growth and DNA replication. It can be used as a multi-functional factor in a variety of antioxidants and can even prevent hormone-induced cancers (Seaton, 2001). In addition, hypoalbuminemia can also reflect systemic inflammation (Jin et al., 2018). Therefore, the LAR can comprehensively reflect the tumor burden, tumor hypoxia, nutritional status, and the systemic inflammation of a patient. In related research reports (Gan et al., 2018; Zheng, Xu & Fu, 2018; Dong & Wang, 2019), the LAR was found to be related to the prognostic survival of patients with nasopharyngeal carcinoma, advanced gastric cancer, liver cancer, and other cancers, and can be an independent poor-prognostic factor. However, the prognostic value of the LAR in colon cancer has not yet been reported in China. To the best of our knowledge, this is the first study to analyze the relationship between the LAR and the prognosis of patients with colon cancer. The results suggest that high preoperative LAR values (LAR > 4.91) can be used as an independent adverse prognosis factor in colon cancer patients.

In this study, the PLR value was associated with the PFS and OS in colon cancer patients in a univariate analysis, while in a multivariate analysis, the PLR value was only negatively correlated with OS, with statistical differences, and the PFS was not statistically significant. The PLR is the ratio of platelets to lymphocytes. Platelets can promote tumor angiogenesis, and tumor cell metastasis (Sabrkhany, Griffioen & Oude Egbrink, 2011) and lymphocytes can exert various “bidirectional” effects, i.e., anti-cancer and cancer-promoting effects, depending on the tumor microenvironment and lymphocyte subtype (Chen et al., 2012; Hiraoka, 2010). Thus, the PLR reflects the activity of platelets in vivo. An increase in the PLR value not only affects the formation of tumor microthrombi, but can also enhance tumor cell deformation and invasive ability, and may even induce tumor-cell infiltration in draining lymph nodes (Wei, Liang & Hong, 2015). Many studies have concluded that a high PLR affects the prognosis of patients with colon cancer, but whether or not it can be used as an independent prognostic factor remains controversial. (Chen, Yao & Liu, 2015). Stated that although the PLR affects the OS and PFS, it is not an independent prognostic factor (Li et al., 2016; Chen, Yao & Liu, 2015). Conversely, You et al. (2016) found that the PLR value is related to OS in both univariate and multivariate analyses, and could be used as an independent prognostic factor for patients with colorectal cancer (Guo et al., 2017; You et al., 2016). The results of this study support the view of You et al., as our data show that PLR has predictive value for establishing the prognosis of OS in colon cancer.

In this study, PNI values and CONUT scores were not correlated with OS and PFS in either a univariate or multivariate analysis. The PNI value is calculated from the peripheral blood albumin and lymphocyte counts, which is an index of the immune and nutritional status of the body. The PNI is normally used to evaluate perioperative risk and nutritional status of patients prior to gastrointestinal surgery (Migita et al., 2013; Onodera, Goseki & Kosaki, 1984). Many recent studies have shown that a low PNI is an independent risk factor of poor prognosis in patients with colorectal cancer (Bailón-Cuadrado et al., 2019; Sasaki et al., 2019). The CONUT score is a newly proposed index to evaluate the immune nutritional status of patients. The CONUT score is obtained by calculating the test values of serum albumin, lymphocyte count and cholesterol concentration. The inclusion of cholesterol concentration is the main difference between the CONUT score and other inflammatory marker scores. Cholesterol, as an important component of the cell membrane, is involved in a variety of signaling pathways related to tumorigenesis, tumor development, and the immune response (Haghikia & Landmesser, 2018; Lyu et al., 2017). There have been many reports indicating that the CONUT score can independently predict the prognosis of a variety of malignant tumors (Toyokawa et al., 2017) and is significantly related to the prognosis of patients with lung cancer, liver cancer, and gastric cancer (Kuroda et al., 2018; Takagi et al., 2017). The PNI and CONUT scores in this study were not related to the prognosis survival of colon cancer patients likely because this was just a small, single-center study.

We compared four indices: the preoperative PLR, PNI, LAR, and CONUT scores. The LAR had the largest area under its ROC curve (0.754), and the sensitivity and specificity of the LAR were also better than those of the PLR and the PNI; these findings indicate that the predictive value of the LAR was optimal. In both a univariate and multivariate analysis, the LAR was significantly correlated with the OS and PFS of patients with colon cancer, and can therefore be used as an independent factor to determine the risk of postoperative recurrence and survival in this group. Overall, the LAR had a stronger effect on the prognosis of colon cancer patients than the PLR, the PNI, and CONUT scores. There are several possible reasons for this finding. First, lymphocyte count is a component of the PLR, the PNI, and CONUT scores, and lymphocytes have a bi-directional effect, i.e., they can promote or inhibit tumors; this is likely to cause instability in these three indices. Second, an increase in the LDH concentration may be more reflective of proliferation, metastasis, and tumor recurrence than changes in other parameters. Third, the LAR is easier to understand, due to its relevance to pathophysiology and its clinical utility, than other measures.

In previous studies, the optimal cut-off value of the LAR to predict survival was: 5.5 for esophageal cancer (Feng et al., 2019), 3.028 (Dong & Wang, 2019) for progressive gastric cancer, 3.59 for nasopharyngeal cancer (Zheng, Xu & Fu, 2018), and in this study, the best cut-off value for determining the prognosis of colon cancer patients was 4.91. Currently, there is no specific standard for the optimal cut-off LAR value, likely because studies of the LAR differ in sample size, screening criteria for selected cases, and optimal cut-off point (with some using patient death as the cut-off point and others using patient disease progression as the cut-off point). These studies may also be impacted by differences in geographic regions in which the studies are conducted.

In examining the relationship between LAR value and clinical pathological characteristics in colon cancer patients, our results showed a correlation between the LAR and CONUT score. This is consistent because while the CONUT score is a nutritional evaluation index, the LAR itself can better reflect the nutritional status of the whole body. We also found that the LAR is associated with M stages, that patients with stage M1a are more likely to experience an increase in LAR values, and that patients with advanced cancer often experience severe long-term nutritional consumption, which in turn can lead to malnutrition and increase inflammation (Fontana et al., 2007). In addition, patients with advanced colon cancer tend to experience high tumor loads, which may lead to the body producing large amounts of circulating inflammatory cytokines causing immunosuppression (Roshani, McCarthy & Hagemann, 2014) making the long-term survival of patients with advanced colon cancer more susceptible to preoperative immune and nutritional status.

In this study, clinical TNM staging was significantly related to OS and PFS in colon cancer patients both in single-factor and multi-factor analyses. TNM staging has been recognized as a helpful criterion for determining prognosis for malignant tumors. Our study reached a similar conclusion with both the LAR value and clinical staging able to evaluate the survival prognosis of colon cancer patients. The results also show that in a single-factor analysis, M-phased and N-phased are significantly correlated with OS and PFS, whereas in a multi-factor analysis, only M-phased is related to OS, which suggests that a single stage factor is unstable in the prognosis evaluation of colon cancer. We regret that the LAR value was not jointly evaluated with TNM in stages to assess colon cancer prognosis. In further studies, our research team will combine the LAR value and TNM stage to predict and analyze the prognosis of colon cancer patients.

In a survival analysis, we found a significant correlation between gender and PFS, with female gender significantly associated with poor prognosis in colon cancer. It is unclear whether the poor prognosis seen in female patients is related to genetic or molecular mechanisms or multiple potential hormones.

In conclusion, we found that the LAR value can be used as an independent adverse prognosis factor in patients with colon cancer, and that this prognosis can lead to multi-dimensional risk stratification for these colon cancer patients. These results suggest that we should carry out effective nutritional support before and after surgery; nutrition support can help reduce the toxic side effects of surgical trauma, postoperative chemotherapy, prolong survival, and improve the quality of life of patients. The results also suggest that the follow-up interval of patients should be adjusted according to LAR value, and the follow-up interval of patients with a high LAR value should be shortened to extend their PFS and OS.

This study has several limitations. First, it was only a single-center data analysis, and the number of patients (126 cases) may be too small to fully represent all the situations of patients with colon cancer. Our research team aims to conduct a multicenter data analysis with a large sample of prospective studies to further validate the role of the LAR in the assessment of colon cancer prognosis. Second, it was a retrospective study with certain inherent limitations. Third, the survival end point was defined as the death of a patient, and thus the results may differ from those studies that use tumor-related death as the survival end point. Fourth, we only evaluated the LAR at a single point in time prior to surgery. Changes in the LAR over the patient’s entire cancer treatment cycle may have a more significant effect on prognosis, but we did not evaluate that in our study.

Supplemental Information

Supplemental Information 1 Raw data.

Click here for additional data file.

Additional Information and Declarations

Competing Interests

Author Contributions

Data Availability

The authors declare that they have no competing interests.

Zhihui Xie performed the experiments, prepared figures and/or tables, authored or reviewed drafts of the paper, and approved the final draft.

Hui Zhou performed the experiments, analyzed the data, authored or reviewed drafts of the paper, and approved the final draft.

Lei Wang conceived and designed the experiments, performed the experiments, prepared figures and/or tables, and approved the final draft.

Yibo Wu conceived and designed the experiments, authored or reviewed drafts of the paper, and approved the final draft.

The following information was supplied regarding data availability:

The raw data is available in the Supplemental File.

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
