# Peer review of "The Significance of the preoperative lactate dehydrogenase/albumin Ratio in the Prognosis of Colon Cancer: a retrospective study"

_PeerJ, doi:10.7717/peerj.13091_

## Round 0.1 · original submission · Major Revisions

I think it will be important to highlight the importance of LAR and lactate dehydrogenase using recommendations from both reviewers.

Reviewer 1 ·

Basic reporting

no comment.

Experimental design

I think it is quite a small number, only 126 cases in 3.5 years of case accumulation. Is this the number of consecutive cases that were included in the study?

Lines 84, 85
I suggest you add "before."
Serum biochemical examination was performed within 1 week before surgery and the data were complete.

All eligible cases are assumed to have undergone radical resection, but can even M1 cases be considered radical surgery?

Are there any errors in the PNI formula? I think this is a very serious error.

Validity of the findings

Lines 127
CONUT's scoring criteria and groupings are in Table 1.

Where are the multivariate results for PFS in N stage and M stage? And what are the multivariate results for OS in N stage?

Does the value of lactate dehydrogenase alone not show a significant difference in PFS or OS? I would like to know.

In other reports, what is the cut-off value for LAR that makes a difference in the prognosis of malignancy? Is there an increase or decrease compared to this study? And why?

Is there a difference between predicting prognosis with LAR and the currently widely used stage? What is the usefulness of preoperative LAR in predicting prognosis? Isn't the stage alone sufficient?

Additional comments

I thought it was very interesting to read that the preoperative lactate dehydrogenase/albumin ratio is useful in predicting the prognosis of colorectal cancer. However, I think some alterations are necessary.

·

Basic reporting

no comment

Experimental design

no comment

Validity of the findings

no comment

Additional comments

see attached PDF

---

## Round 0.2 · Minor Revisions

Dear Dr. Wu

There is a very minor revision needed before accepting your manuscript for publication in PeerJ. Please see reviewer 1's comments and provide data as requested. I think this will improve the manuscript.

Reviewer 1 ·

Basic reporting

nothing

Experimental design

nothing

Validity of the findings

Thank you for your sincere answer to my question. I have a further question for your answer.

1. In general, factors that show significant differences in univariate analysis are examined in multivariate analysis and the results are presented. Even if there is no significant difference, the results of multivariate analysis for PFS in stage N and M , and the results of multivariate analysis for OS in stage N should be presented.
2. If lactate dehydrogenase alone is significantly correlated with OS and PFS in colorectal cancer patients, I think it should be included as a factor to be evaluated. Lactate dehydrogenase alone is not significant compared to LAR?

Additional comments

nothing

---

## Round 0.3 · accepted · Accept

I am glad to accept this manuscript.